# Rewiring of endogenous signaling pathways to genomic targets for therapeutic cell reprogramming

Krzysztof Krawczyk [1], Leo Scheller [1], Hyojin Kim[1] & Martin Fussenegger [1,2]*

Rewiring cellular sensors to trigger non-natural responses is fundamental for therapeutic cell engineering. Current designs rely on engineered receptors that are limited to single inputs, and often suffer from high leakiness and low fold induction. Here, we present Generalized Engineered Activation Regulators (GEARs) that overcome these limitations by being pathway-specific rather than input-specific. GEARs consist of the MS2 bacteriophage coat protein fused to regulatory or transactivation domains, and work by rerouting activation of the NFAT, NFκB, MAPK or SMAD pathways to dCas9-directed gene expression from genomic loci. This system enables membrane depolarization-induced activation of insulin expression in β-mimetic cells and IL-12 expression in activated Jurkat cells, as well as IL-12 production in response to the immunomodulatory cytokines TGFβ and TNFα in HEK293T cells. Engineered cells with the ability to reinterpret the extracellular milieu have potential for applications in immunotherapy and in the treatment of metabolic diseases.

---

[1] Department of Biosystems Science and Engineering, ETH Zurich, Mattenstrasse 26, CH-4058 Basel, Switzerland. [2] Faculty of Science, University of Basel, Mattenstrasse 26, CH-4058 Basel, Switzerland. *email: fussenegger@bsse.ethz.ch

Numerous active clinical trials are investigating the therapeutic potential of engineered cell therapies, mainly in oncology and organ regeneration[1]. Chimeric antigen receptor T (CAR-T) cells were successfully applied for killing lymphoma cells and have been FDA-approved[2]. Current research for enhancing CAR-T cell function is mainly focused on improving synthetic receptors and on conditional expression of immunostimulatory cytokines, such as interleukin 2 (IL-2) and interleukin 12 (IL-12), to combat solid tumors more effectively[3–6]. Besides immunotherapy, engineered cells have been applied in many other proof-of-concept treatment strategies in animal disease models, including the treatment of metabolic diseases using encapsulated cells[7,8], and therapeutic agent delivery using transgenic mesenchymal stem cells[9,10]. However, current designs of engineered cells are still limited by the lack of efficient and versatile molecular devices. New methods for precise control of cellular-signaling pathways are needed to further improve cell therapies[6,11–16].

Recent efforts have led to the development of synthetic receptors for endogenous promoter activation in response to vascular endothelial growth factor (VEGF), which is important for tumor growth and immunoevasion[17,18]. However, these systems do not respond to inducer concentrations that typically activate native receptors, and are excluded from natural control mechanisms. Natural signal transduction relies on a multitude of signaling pathways that have evolved for efficient and tightly regulated dynamic behavior. Additionally, many signaling pathways interact with each other, forming multidimensional networks that integrate various inputs to generate sophisticated situation-dependent responses. Therefore, precise quantitative and temporal regulation is critical for obtaining desired effects[19]. Rewiring natural receptors to specific transgene expression has been established for several proof-of-concept cellular therapies[8,20]. However, rewiring these pathways directly to endogenous targets remains a challenge.

Here, we present a modular system for the direct repurposing of endogenous signaling pathways to activate native or synthetic promoters. This system relies on molecular devices called Generalized Engineered Activation Regulators (GEARs). Each GEAR consists of a MS2 bacteriophage coat protein (MCP) fused to natural regulatory or transactivation domains of key signaling pathways. GEARs combined with catalytically dead CRISPR-associated protein 9 (dCas9) and a synthetic guide RNA (sgRNA) containing two MS2 coat protein-binding loops (MS2)[21] (Fig. 1a) can be used to reroute calcium signaling, the TGFβ/SMAD pathway, the NFκB pathway, and the MAPK/ERK pathway (Fig. 1b). The proof-of-concept applications include membrane depolarization-induced activation of insulin expression in β-mimetic cells, IL-12 expression in activated immortalized human T-lymphocytes (Jurkat), and activation of IL-12 production in response to immunosuppressive TGFβ or immunostimulatory TNFα in HEK293T cells. GEARs have potential for applications in therapeutic cell engineering, especially in immunotherapy and in the treatment of metabolic diseases.

## Results

**GEAR_NFAT enables rewiring of intracellular calcium signaling.** Calcium is one of the most pivotal second messengers, intersecting with numerous signaling pathways to mediate multiple cellular processes[22]. Therefore, we first focused on nuclear factor of activated T-cells (NFAT), which is dephosphorylated upon activation of a calcium-dependent calmodulin–calcineurin cascade, and designed $GEAR_{NFAT}$, which consists of MCP fused to the transactivation domains of p65 and HSF1 ($p65_{TA}$–$HSF1_{TA}$), and the regulatory domain of NFAT ($NFAT_{reg}$). $GEAR_{NFAT}$ translocates to the nucleus (Supplementary Fig. 1) and activates sgRNA-specified gene expression in response to an increase of the intracellular calcium level. We used $GEAR_{NFAT}$ to upregulate insulin expression upon cell-membrane depolarization in HEK293T cells, mimicking the response observed in pancreatic β-cells[23] (Fig. 2a) and evaluated the system performance by means of a reporter gene assay. HEK293T cells were engineered to express L-type voltage-gated calcium channel (LVGCC) in order to enhance membrane depolarization-driven calcium influx[8]. Incorporating $GEAR_{NFAT}$ into these cells enabled membrane depolarization-dependent expression of a reporter gene controlled by an insulin promoter (Fig. 2b). $GEAR_{NFAT}$ activation led to significant reporter gene expression within 8 h, reaching a maximum after 36 h (Supplementary Fig. 2). Furthermore, when expressed in a β-mimetic cell line[8], $GEAR_{NFAT}$ increased endogenous insulin gene expression (Fig. 3a, b) and protein production (Fig. 3c) in response to membrane depolarization.

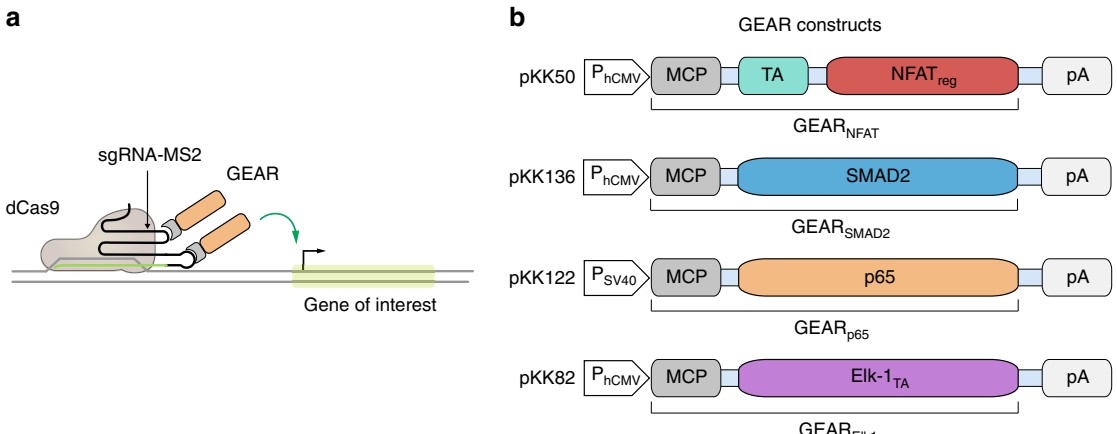

**Fig. 1 GEAR toolbox. a** Catalytically inactive Cas9 (dCas9) binds a DNA sequence complementary to single-stranded guide RNAs equipped with two MS2 aptamers (sgRNA-MS2). GEARs can bind the MS2 aptamers via MS2 coat protein (MCP) domains and activate gene expression in response to endogenous signaling. **b** All GEAR variants contain MCP at the N-terminal end. $GEAR_{NFAT}$ contains a transactivation domain (TA) and the regulatory domain of NFAT ($NFAT_{reg}$). Plasmid pKK50 encodes $GEAR_{NFAT}$ under control of human cytomegalovirus promoter ($P_{hCMV}$-$GEAR_{NFAT}$-pA). $GEAR_{SMAD2}$ contains full-length SMAD2. Plasmid pKK136 encodes $GEAR_{SMAD2}$ under control of human cytomegalovirus promoter ($P_{hCMV}$-$GEAR_{SMAD2}$-pA). $GEAR_{p65}$ contains NFκB transcription regulator subunit p65. Plasmid pKK122 encodes $GEAR_{p65}$ under control of human cytomegalovirus promoter ($P_{hCMV}$-$GEAR_{p65}$-pA). $GEAR_{Elk}$ contains the transactivation domain of Elk1. Plasmid pKK82 encodes $GEAR_{Elk}$ under control of human cytomegalovirus promoter ($P_{hCMV}$-$GEAR_{Elk}$-pA). pA is a polyadenylation signal.

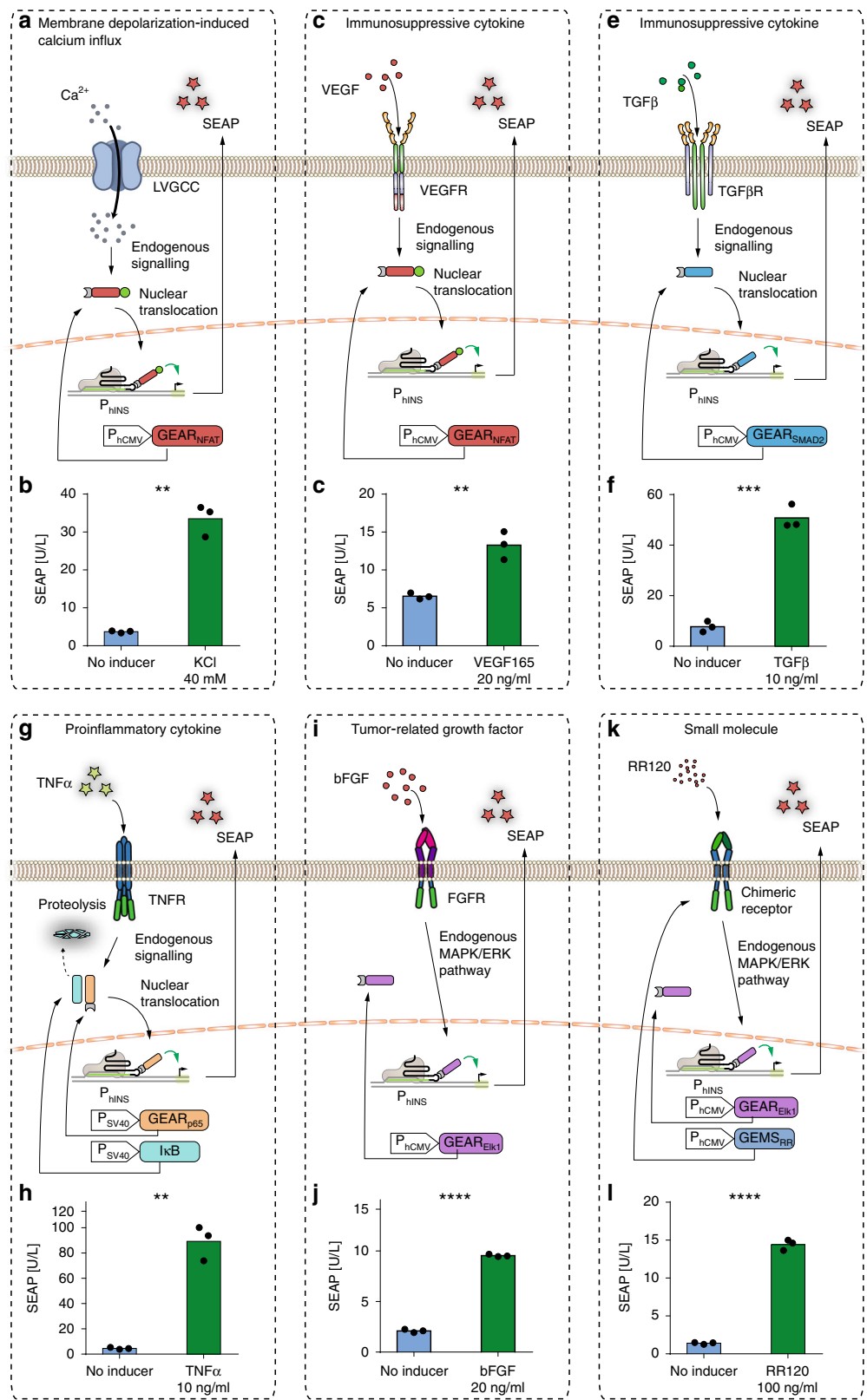

Next, we tested the multiplexing capability of GEARs and simultaneously transfected dCas9, GEAR$_{NFAT}$, and sgRNAs in order to upregulate gene expression from the native genomic loci of insulin and IL-12. The gene expression levels of both insulin and IL-12 were highly upregulated upon cell stimulation with KCl (Fig. 3d).

VEGF signaling is another example of an important pathway that is responsive to transient increases of cytosolic calcium concentration. Therefore, we speculated that stimulation of the VEGF receptor (VEGFR) could also be used to activate GEAR$_{NFAT}$. We found that GEAR$_{NFAT}$ was functional in VEGFR-expressing HEK293T cells and led to robust transgene expression (Fig. 2c, d), as well as endogenous gene expression (Supplementary Fig. 3a) in response to stimulation with VEGF. Similarly, rewiring of VEGFR using GEAR$_{NFAT}$ was functional in

**Fig. 2 GEAR-driven transgene expression.** HEK293T cells containing a reporter plasmid for human insulin promoter ($P_{hINS}$) and expressing GEARs, dCas9, and a human insulin promoter-specific sgRNA-MS2. At 24 h after the beginning of transfection, cells were stimulated for 36 h with the indicated inducer and then the reporter protein SEAP was quantified in the cell culture supernatant. **a, b** Membrane depolarization-dependent $GEAR_{NFAT}$ activation. Cells expressing $GEAR_{NFAT}$ and the L-type voltage-gated calcium channel were depolarized with 40 mM potassium chloride (KCl). **c, d** $GEAR_{NFAT}$ activation upon stimulation with vascular endothelial growth factor 165 (VEGF165). Cells expressing $GEAR_{NFAT}$ and VEGF receptor 1 (VEGFR1) were stimulated with 20 ng/mL VEGF165. **e, f** Transforming growth factor β (TGFβ)-induced $GEAR_{SMAD2}$ activation. Cells expressing $GEAR_{SMAD2}$ and TGFβ receptor 2 (TBRII) were stimulated with 10 ng/mL TGFβ. **g, h** $GEAR_{p65}$ activation upon stimulation with tumor necrosis factor α (TNFα). Cells transfected with $GEAR_{p65}$ and IκB-encoding plasmids in a 1:1 ratio were stimulated with 10 ng/mL TNFα. **i, j** $GEAR_{Elk1}$ activation upon basic fibroblast growth factor (bFGF) stimulation. Cells expressing $GEAR_{Elk1}$ were stimulated with 20 ng/mL bFGF. **k, l** $GEAR_{Elk}$ activation upon chimeric receptor stimulation. Cells expressing $GEAR_{Elk1}$ and MAPK-GEMS were stimulated with the synthetic azo-dye RR120 (100 ng/mL). Green bars represent SEAP concentrations measured in the supernatant of stimulated cells (mean values). Blue bars represent controls without inducer. Black dots correspond to individual data points of $n = 3$ biological replicates. $^{**}p < 0.01$, $^{***}p < 0.001$, $^{****}p < 0.0001$. Statistical significance was calculated using a two-tailed $t$-test. A detailed description of the statistical analysis is provided in Supplementary Table 5. Source data are provided as a Source Data file.

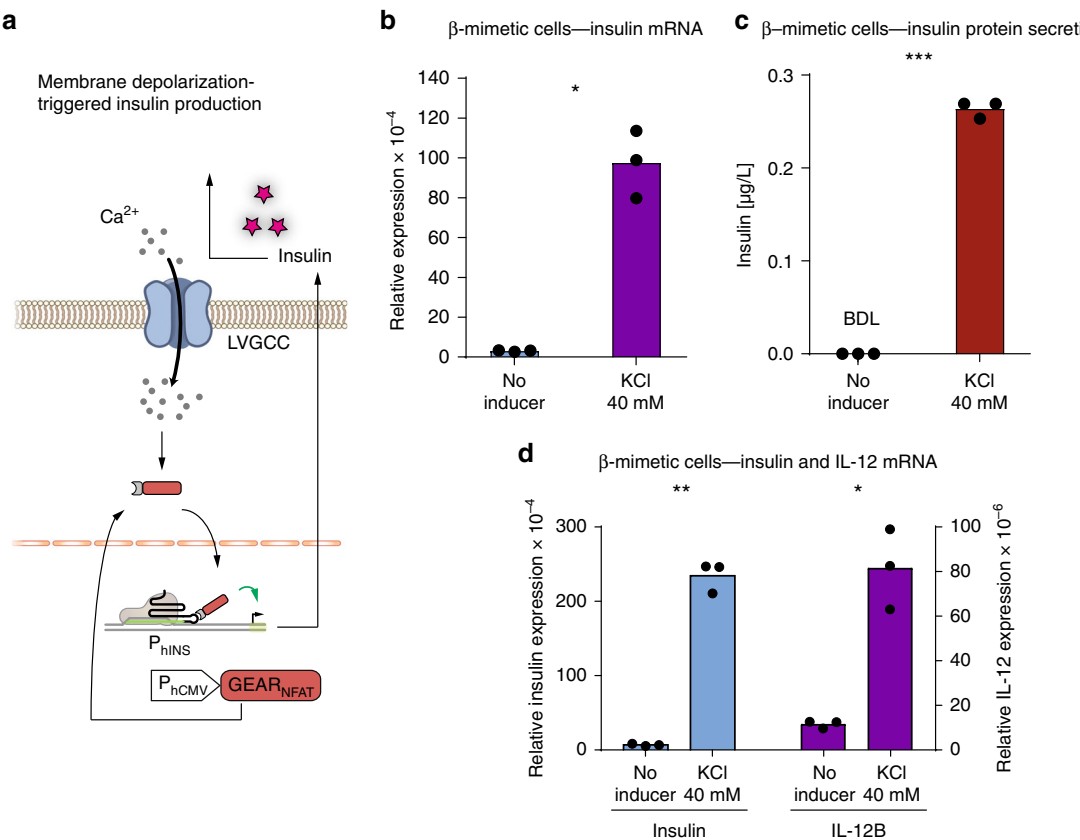

**Fig. 3 Rewiring intracellular calcium signaling to genomic targets. a** Membrane depolarization-triggered insulin expression. Cell membrane depolarization causes calcium influx via L-type voltage-gated calcium channels, followed by nuclear translocation of $GEAR_{NFAT}$, and endogenous insulin promoter ($P_{hINS}$) activation. **b, c** 24 h after the beginning of transfection, β-mimetic cells expressing dCas9, $GEAR_{NFAT}$, and a human insulin promoter-specific sgRNA were depolarized with 40 mM KCl. Insulin mRNA **b** and protein in the supernatant **c** were quantified after 36 h. **d** $GEAR_{NFAT}$-driven multiple gene activation. β-Mimetic cells expressing dCas9, L-type voltage-gated calcium channel, $GEAR_{NFAT}$ and $P_{hINS}$-specific sgRNA, as well as $P_{hIL-12B}$-specific sgRNA were depolarized with 40 mM KCl. Insulin and IL-12B mRNA levels were quantified after 36 h. Blue bars and the left axis correspond to insulin mRNA. Violet bars and the right axis correspond to IL-12B mRNA. Black dots correspond to individual data points of $n = 3$ biologically independent samples. BDL below detection limit, ns nonsignificant, $^{*}p < 0.05$, $^{**}p < 0.01$, $^{***}p < 0.001$. Statistical significance was calculated using a two-tailed $t$-test. A detailed description of the statistical analysis is provided in Supplementary Table 5. Source data are provided as a Source Data file.

human mesenchymal stem cells (hMSCs-TERT) (Supplementary Fig. 4a). This response uses the naturally optimized behavior of native receptors and endogenous signaling to achieve the high sensitivity that is observed in nature. Direct comparison of the VEGFR/$GEAR_{NFAT}$ system with previously published synthetic receptors (MESA)[18] showed the superiority of $GEAR_{NFAT}$ under the tested conditions (Supplementary Fig. 5).

NFAT was originally discovered as a mediator of the responses of activated T-cells[24]. In the native pathway, stimulation of the T-cell receptor complex leads to increased cytosolic calcium concentration, followed by the activation of T-cell responses. Thus, expressing $GEAR_{NFAT}$ allowed us to hijack calcium signaling in a T-cell line (Jurkat) to increase the expression of endogenous immunostimulatory cytokine IL-12[25] (Fig. 4a), thereby illustrating the potential value of GEARs in the context of immunotherapy. We confirmed that $GEAR_{NFAT}$ mediated upregulation of IL-12 at both the mRNA and the protein level in response to increased cytosolic calcium concentration (Fig. 4b, c).

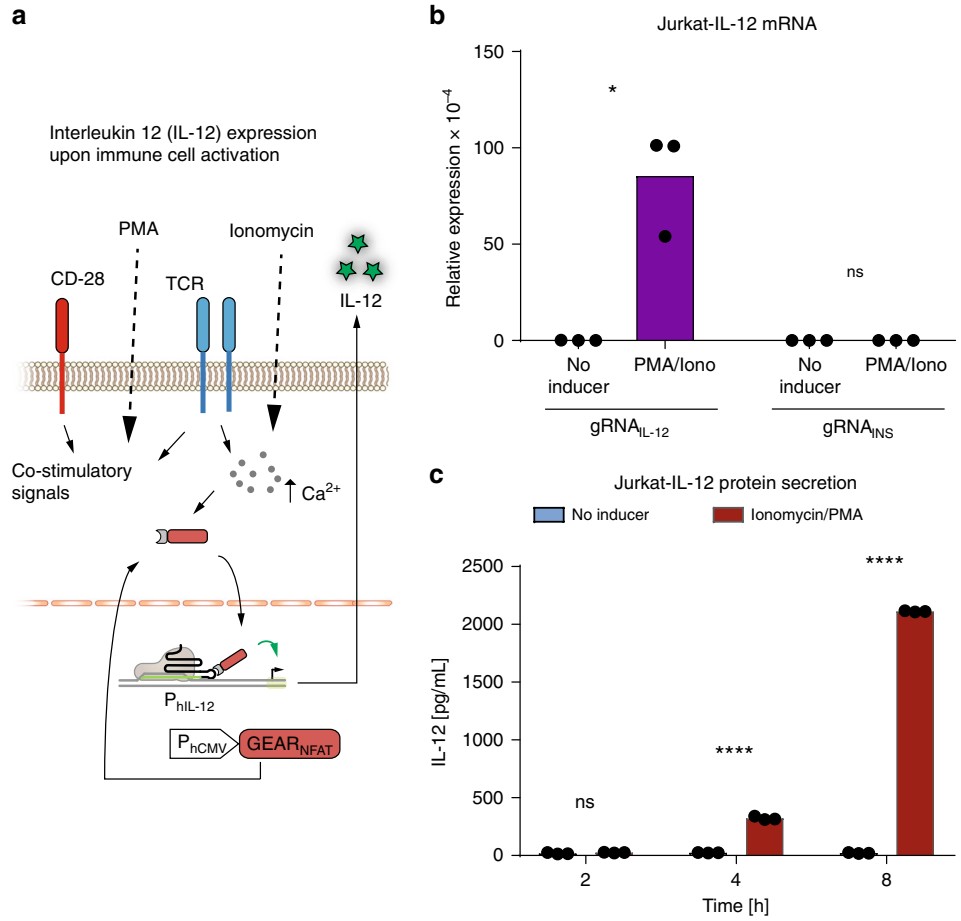

**Fig. 4 Interleukin 12 expression upon T-cell activation. a** Schematic representation of T-cell activation. T-cell receptor (TCR) signaling is transduced via calcium. An increase in the intracellular calcium level activates GEAR$_{NFAT}$ and leads to elevated expression of interleukin 12 (IL-12). **b, c** Jurkat cells stably expressing dCas9, GEAR$_{NFAT}$, and a human IL-12B promoter-specific sgRNA, or an insulin promoter-specific sgRNA as the negative control, were activated with 2 µg/mL ionomycin and 20 ng/mL PMA. **b** IL-12B mRNA was quantified after 8 h. **c** IL-12B protein was quantified from the cell culture supernatant after 2, 4, and 8 h. Black dots correspond to individual data points of $n = 3$ biological replicates. ns nonsignificant, **$p < 0.01$, ****$p < 0.0001$. Statistical significance was calculated using a two-tailed $t$-test. A detailed description of the statistical analysis is provided in Supplementary Table 5. Source data are provided as a Source Data file.

We next examined whether GEAR$_{NFAT}$ influences the expression levels of native NFAT-dependent genes by means of a reporter gene assay, as well as by qPCR analysis of known NFAT-responsive genes. We found that GEAR$_{NFAT}$ downregulated expression of NFAT-driven genes in a GEAR$_{NFAT}$-positive stable cell line of Jurkat cells (Supplementary Fig. 6), as well as in HEK293T cells transfected with a P$_{NFAT}$-driven reporter plasmid (Supplementary Fig. 7). The downregulation of NFAT-driven genes was dependent on the amount of the transfected GEAR$_{NFAT}$-encoding plasmid: increasing the amount of the plasmid increases production of the GEAR-controlled reporter protein (SEAP), while decreasing the production of P$_{NFAT}$-driven NanoLuc luciferase (Supplementary Fig. 7).

In the design of GEARs, high flexibility and modularity are achieved by the physical separation of a regulatory module and the dCas9-dependent DNA-binding module. For the regulatory module, we took advantage of the small size of the MCP protein, because we speculated that a smaller fusion protein might be more efficiently regulated by the cellular machinery, e.g. for induced nuclear translocation. In a direct comparison of GEAR$_{NFAT}$ and a construct composed of dCas9 fused to the NFAT$_{reg}$ and a transactivator (named CaRROT)[26], GEAR$_{NFAT}$ outperformed CaRROT under the tested conditions (Supplementary Figs. 2a and 8).

**Generalizing functionality of GEARs to other signaling pathways.** We next generalized the functional principle of GEAR$_{NFAT}$ to incorporate other signaling pathways. We engineered GEAR$_{SMAD2}$ to transduce signals from receptors of the transforming growth factor beta (TGFβ) superfamily. The functionality of the construct was confirmed with P$_{hINS}$-driven reporter gene expression in response to stimulation with TGFβ in HEK293T cells (Fig. 2e, f) and hMSCT-TERT cells (Supplementary Fig. 4b) transiently expressing the TGFβ receptor, as well as with activation of the endogenous insulin promoter (Supplementary Fig. 3b). As proof of concept for converting the normally immunosuppressive effect of TGFβ into an immunostimulatory response, TGFβ signaling was rewired to the expression of IL-12 (Fig. 5a, b).

Next, we generated the NFκB-dependent GEAR (GEAR$_{p65}$) by fusing MCP to the NFκB subunit p65. NFκB nuclear translocation depends on the degradation of its binding partner IκB, which exposes the nuclear localization signal (NLS) of p65. GEAR$_{p65}$ was therefore coexpressed with IκB to avoid non-induced p65 activity. We used GEAR$_{p65}$ to redirect the signal of TNFα. Stimulation with TNFα leads to the activation of endogenously expressed receptors and increased P$_{hINS}$-driven transgene activation in HEK293T (Fig. 2g, h) and hMSC-TERT (Supplementary Fig. 4c). GEAR$_{p65}$ targeted for P$_{hINS}$ (Supplementary Fig. 3c) or P$_{hIL12}$ increased

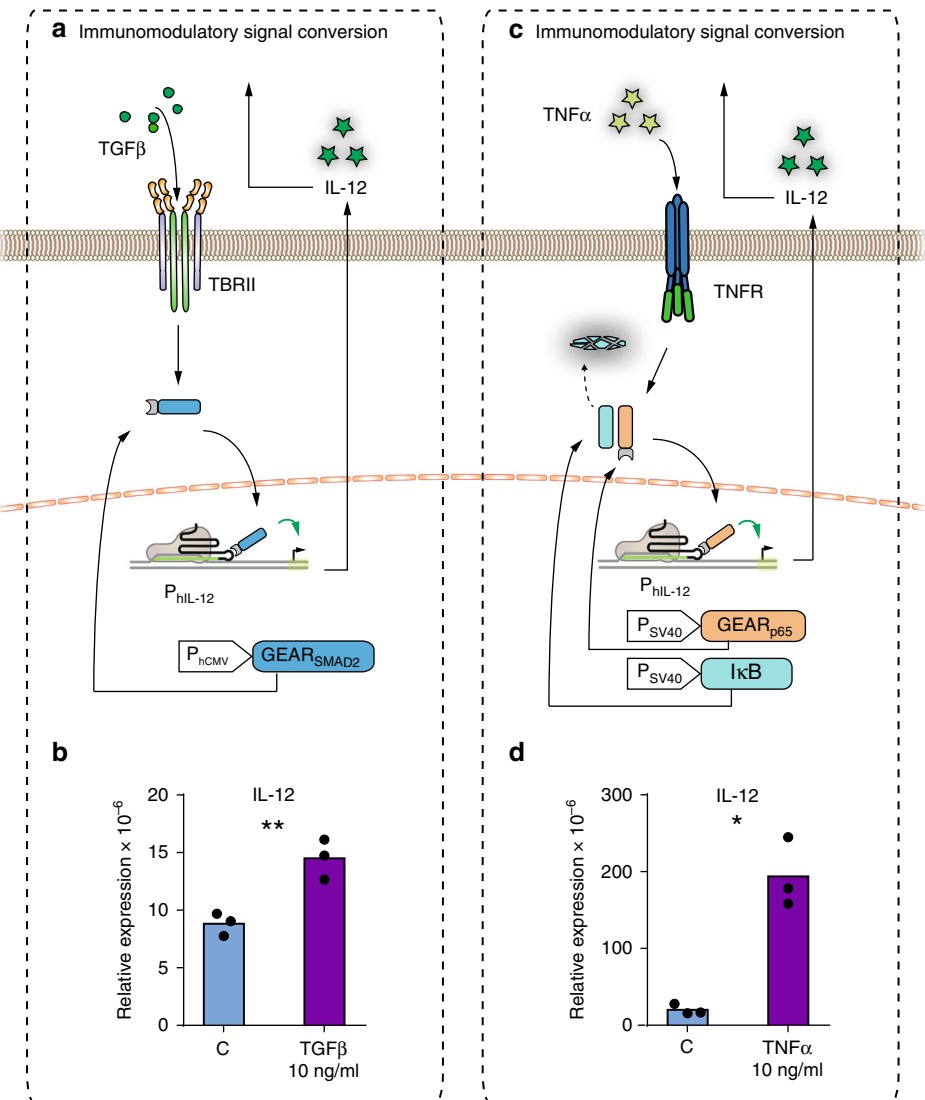

**Fig. 5 Rewiring inflammation-related signaling pathways to genomic targets. a** Immunomodulatory signal conversion. TGFβ-induced TBRII activation causes nuclear translocation of GEAR$_{SMAD2}$ and subsequent IL-12 promoter (P$_{hIL-12}$) activation. **b** At 24 h after the beginning of transfection, HEK293T cells expressing dCas9, GEAR$_{SMAD2}$, TBRII, and a human IL-12B promoter-specific sgRNA were stimulated with 10 ng/mL TGFβ. IL-12B mRNA was quantified after 36 h. **c** Immunomodulatory signal switch. TNFα-induced TNFR activation causes IκB degradation, resulting in nuclear translocation of GEAR$_{p65}$ and IL-12 promoter (P$_{hIL-12}$) activation. **d** At 24 h after the beginning of transfection, HEK293T cells expressing dCas9, GEAR$_{p65}$, IκB, and a human IL-12B promoter-specific sgRNA were stimulated with 10 ng/mL TNFα. IL-12B mRNA was quantified after 36 h. Violet bars represent mRNA expression (relative to GAPDH) in stimulated cells (mean value). Blue bars represent mRNA expression in controls without inducer. Black dots correspond to individual data points of $n = 3$ biological replicates. *$p < 0.05$, **$p < 0.01$. Statistical significance was calculated using a two-tailed $t$-test. A detailed description of the statistical analysis is provided in Supplementary Table 5. Source data are provided as a Source Data file.

endogenous insulin or IL-12 expression, respectively (Fig. 5c, d). This effect might be relevant for immunotherapy, as IL-12 is an important immunostimulatory cytokine, but its expression is inhibited upon exposure to TNFα in some cell types[27,28].

MEK/ERK-activated GEAR (GEAR$_{Elk1}$) consists of MCP fused to the Elk1 transactivation domain (Elk1$_{TA}$). GEAR$_{Elk}$ (P$_{hCMV}$-GEAR$_{Elk1}$-pA, pKK82) was used to rewire signaling of fibroblast growth factor (FGF), an important signaling molecule over-expressed in certain tumors[29]. Incubation with basic FGF (bFGF) leads to increased transgene expression in HEK293T (Fig. 2i, j) and hMSC-TERT (Supplementary Fig. 4d). Moreover, the system is compatible with the recently developed generalized extra-cellular molecule sensor (GEMS)[30] platform (Fig. 2k, l). However, GEAR$_{Elk1}$ did not increase expression of tested endogenous genes: insulin, IL-12, or IL-2.

**Functional characterization and non-specific effects.** Control experiments conducted without GEARs or with unspecific sgRNAs showed little or no increase in reporter gene expression, or in endogenous gene activation that could be attributed to the native host transcription machinery, while expression of GEARs together with specific sgRNAs also increased non-induced reporter gene expression (Supplementary Figs. 9 and 10). We further evaluated the effect on the expression of endogenous genes, focusing on insulin as a gene typically not expressed in HEK293T cells, as well as TGFβ, which is highly expressed in this cell type. The insulin transcript was not detected in the absence of GEARs. The transcription level of TGFβ did not show any major change due to binding the non-activating dCas9 complex (Supplementary Fig. 11). Dose-response experiments performed for VEGF, TGFβ, TNFα, and bFGF (using systems incorporating

GEAR$_{NFAT}$, GEAR$_{SMAD2}$, GEAR$_{p65}$, and GEAR$_{Elk1}$, respectively) confirmed that the sensitivity of GEAR-based systems depends on the sensitivity of the receptor (Supplementary Fig. 12). Their kinetic profiles showed that activation was observable after 4 h (Supplementary Fig. 2). Since reporter gene assay results (Fig. 2) do not always correlate well to RNA content quantified by qPCR (Fig. 3), we performed both assays. These assays showed qualitatively similar increases in SEAP reporter gene activation at the mRNA and protein levels (Supplementary Fig. 13).

## Discussion

Therapeutic cell engineering requires various molecular devices and methods to achieve the desired cell behavior. Many approaches involve regulating transcription by employing orthogonal[31,32] or native[8,33,34] signaling pathways, or combinations of the two[8,35]. Native cell signaling pathways are optimized by evolution to precisely regulate cellular behavior. Hijacking a signaling node that is shared by multiple pathways thus enables us to add a novel function to an operational system, such as expression of IL-12 concomitantly with T-cell line activation. Hence, GEARs both complement and extend the common trend in synthetic biology to design inducer-specific systems[36].

Development of dCas9 greatly facilitated the conditional modulation of endogenous gene expression. Traditional approaches, which involve inducible expression of the dCas9 protein, typically require more than 2 days from the activation time until the appearance of a detectable effect, and the effect lasts for hours to days[37–40]. The dynamics is slowed down by the need for a two-step transcriptional control system, consisting of transcription of dCas9 followed by transcription of target genes, as well as the stability (in the timescale of hours) of dCas9 mRNA, which slows the off-kinetics. GEAR components can bypass both delays by eliminating the requirement of inducible dCas9 expression, and by following the dynamics of the endogenous signaling cascade.

The therapeutic effect of engineered cells often depends on the function of the whole cell, rather than on the expression of a single gene. Jurkat cells with stable incorporation of GEAR$_{NFAT}$ maintain the expression of native NFAT-dependent genes related to cell activation, as shown by the upregulation of IL-2 and other tested genes. In general, gene expression levels are controlled by numerous factors, which include the promoter sequence, the epigenetic state of chromatin, and the activity of related signaling pathways. Typically, linking a signaling pathway to transgene expression increases its basal level, even in the non-activated state[31,33,34]. The magnitude of that effect depends on the signaling pathway, as it was observed for GEAR-driven reporter gene expression (time-dependent increase visible in Supplementary Fig. 2), as well as on the expression strength of the GEAR (Supplementary Fig. 7a). Simultaneously, GEARs sequester parts of the transcription machinery, leading to decreased expression of genes controlled by natural transcription factors (Supplementary Figs. 6 and 7b). dCas9-based activation of genomic targets that are additionally regulated by epigenetics is typically context-dependent and can be affected by the targeted region, as well as by the choice of transactivators[21,41]. The dCas9 complex might interfere with natural transcription factor–DNA interactions by steric hindrance, leading to either inhibition or activation of transcription[42–44], which opens further possibilities to tune the system. As with other DNA-targeting gene switches, there could be possible off-target effects[45] that could be minimized by the careful choice of the targeted region, the signaling pathway and the GEAR expression level. Rerouting or tapping into endogenous signaling pathways may provide engineering opportunities to create designer cells and adapt their behavior for therapeutic purposes.

Synthetic receptors are another type of molecular devices that are currently used in therapeutic cell engineering. They can sense a soluble input and transduce the signal either by viral protease-mediated release of an effector protein, or by activating native signaling pathways. Typically, TEV protease-based systems require high inducer concentrations to trigger receptor dimerization and thus may not interface optimally with natural control mechanisms. GEARs, on the other hand, respond to inducer concentrations in the same range as native receptors. In a direct comparison of GEAR$_{NFAT}$-driven gene expression in response to VEGF receptor activation with the TEV protease-based synthetic MESA receptor system[18] GEAR$_{NFAT}$ showed superior performance in terms of achieved fold induction and sensitivity. The second class of synthetic receptors, which includes GEMS[30], relies on native signaling pathways. Hence, GEARs can be combined with GEMS to form artificial signaling pathways for normally non-signaling molecules.

We believe GEARs have great potential to extend the functionality of therapeutic cells, and could also find a variety of applications in basic research, especially in studies of cell signaling and systems biology.

## Methods

**Genetic constructs used in this study**. Comprehensive design and construction details for all original expression vectors are provided in Supplementary Table 1. Targeting sequences of sgRNAs are listed in Supplementary Table 2.

**Guide RNA design**. sgRNA target sequences were designed using Benchling (Benchling [Biology Software]. (2019). Retrieved from [https://benchling.com]). This software calculates on-target and off-target scores based on algorithms developed by Doench et al. [46] and Hsu et al. [47].

**Cell culture and transfection**. Human embryonic kidney cells (HEK293T, ATCC: CRL-11268) were obtained from ATCC and cultivated in Dulbecco's modified Eagle's medium (DMEM; cat. no. 52100-039; Thermo Fischer Scientific, Waltham, MA, USA) supplemented with 10% fetal bovine serum (FBS; cat. no. F7524, lot no. 022M3395, Sigma-Aldrich), 100 U/mL penicillin and 100 μg/mL streptomycin (penicillin–streptomycin solution 100×; cat. no. L0022, Biowest, Nuaillé, France) and grown at 37 °C in a humidified atmosphere containing 5% CO$_2$. For transfection, 35,000 cells were seeded per cm$^2$ of the cell culture dish and, after 24 h, incubated for another 6 h with a 1:6 DNA:PEI (Polyethylenimine MAX; MW 40,000, cat. no. 24765-2; Polysciences Inc., Warrington, PA, USA) solution containing 1.5 μg DNA per cm$^2$ of transfected cells. Immortalized human mesenchymal stem cells (hMSC-TERT) were a kind gift from Moustapha Kassem (University Hospital of Aarhus and University Hospital of Odense, Denmark). They were cultivated and transfected the same way as HEK-293T. Jurkat cells (Jurkat, Clone E61, ATCC: TIB152™) were a kind gift from Sai Reddy (ETHZ, Switzerland). Jurkat cells were cultivated in RPMI medium (cat. no. 72400-021; Thermo Fischer Scientific, Waltham, MA, USA) supplemented with 10% fetal bovine serum 100 U/mL penicillin and 100 μg/mL streptomycin. Transfection was performed using Xfect Transfection Reagent (cat. no. 631318; Takara Bio Europe SAS, St. Germain en Laye, France) according to the manufacturer's instructions. Briefly, 2 million cells were resuspended in 2 mL of fresh media and seeded into a 35 mm dish. 5 μg of plasmid DNA was mixed with 1.5 μL of Xfect reagent, incubated for 10 min and added to the cells. Detailed transfection protocols are provided in Supplementary Table 3. None of the cell lines used in this study is listed in the Register of Misidentified Cell Lines, ICLAC. Cell lines were authenticated by means of microscopy and by activation-dependent IL-2 expression for Jurkat cells. Cells were not tested for mycoplasma contamination.

**Stable cell lines**. Jurkat cells were transfected with plasmids encoding SB100X expression vector pCMV-T7-SB100 (P$_{hCMV}$-SB100X-pA) and the SB100X-specific transposon-containing plasmids pKK171 (ITR-P$_{hCMV}$-dCas9-pA:P$_{RPBSA}$-dTomato-P2A-BlastR-pA-ITR) and pKK172 (ITR-P$_{hCMV}$-GEAR$_{NFAT}$-pA:P$_{RPBSA}$-dTomato-P2A-BlastR-pA-ITR). After 48 h, cells were selected with conditioned medium containing 5 μg/mL blasticidin for 21 days. Conditioned medium was prepared by sterile filtration of cell culture medium from wild-type Jurkat cells after 48 h in culture. Next, blasticidin-resistant cells were transfected with pCMV-T7-SB100 and pLeo1209 (ITR-P$_{hU6}$-sgRNA$_{IL-12}$-pA-ITR:P$_{RPBSA}$-BFP-P2A-PuroR-pA-ITR) and 48 h later, selected with conditioned medium containing 0.5 μg/mL puromycin for 21 days. Stably transgenic puromycin-resistant cell populations were selected by FACS single-cell sorting based on cellular fluorescence relative to wild-type Jurkat (BFP: 405 nm laser, 450/50 bandpass filter; dTomato: 561 nm laser,

570 nm long-pass filter, 586/15 bandpass filter) using a Becton Dickinson LSRII Fortessa flow cytometer (Becton Dickinson, Allschwil, Switzerland).

**Cell stimulation**. Transfected HEK293T cells were washed with DMEM containing 1% FBS, 100 U/mL penicillin, and 100 µg/mL streptomycin for 18 h, and incubated with the concentrations indicated in the respective figures for 36 h (unless stated otherwise) with one of the following reagents: recombinant human VEGF165 (cat. no. 100-20, Peprotech, London, UK), recombinant human FGF-basic (154 a.a.) (cat. no. 100-18B, Peprotech, London, UK), recombinant human TNFα (cat. no. 300-01A, Peprotech, London, UK), or recombinant human TGF-β1 (cat. no. 100-21, Peprotech, London, UK). Transfected hMSC-TERT cells were stimulated in DMEM containing 10% FBS, 100 U/mL penicillin, and 100 µg/mL streptomycin. Jurkat cells were centrifuged and resuspended in fresh growth medium containing the indicated concentrations of ionomycin (cat. no. I0634-1MG, Sigma-Aldrich) and phorbol 12-myristate 13-acetate (PMA; cat. no. P1585-1MG, Sigma-Aldrich).

**Analytical assays**. SEAP (human placental secreted alkaline phosphatase) levels were profiled in cell culture supernatants using a colorimetric assay. 100 µL 2x SEAP assay buffer (20 mM homoarginine, 1 mM MgCl$_2$, 21% diethanolamine, pH 9.8) was mixed with 80 µL heat-inactivated (30 min at 65 °C) cell culture supernatant. After the addition of 20 µL substrate solution (120 mM $p$-nitrophenyl phosphate; cat. no. AC128860100, Thermo Fisher Scientific, Waltham, MA, USA), the absorbance time course was recorded at 405 nm and 37 °C using a Tecan Genios PRO plate reader (cat. no. P97084; Tecan Group AG, Maennedorf, Switzerland) and the SEAP levels were determined as follows: first, absorbance change over time (slope) was calculated. According to the Beer–Lambert's law, absorbance is proportional to the concentration of a colored compound and depends on the light path length ($d$) and extinction coefficient ($\varepsilon$) ($\varepsilon$ for $p$-nitrophenyl ($\varepsilon_{pNP}$) = 18.600 M$^{-1}$ cm$^{-1}$). Enzymatic activity EA [U/L] was calculated from the equation: EA = slope × dilution factor × $\varepsilon_{pNP}^{-1}$ × $d^{-1}$ [48].

NanoLuc® luciferase was quantified in cell culture supernatants using the Nano-Glo® Luciferase Assay System (cat. no. N1110; Promega, Duebendorf, Switzerland). In brief, 7.5 µL of cell culture supernatant was added per well of a black 384-well plate and mixed with 7.5 µL substrate-containing assay buffer. Total luminescence was quantified using a Tecan Genios PRO plate reader (Tecan Group AG).

**ELISA**. Human Insulin was quantified with the Mercodia Insulin ELISA (cat. no. 10-1113-01, Mercodia, Uppsala, Sweden). IL-12 was quantified with the human IL-12 (p40) ELISA Kit (cat. no. KAC1561, Thermo Fisher Scientific, Waltham, MA, USA).

**Quantitative RT–PCR**. Total RNA of HEK293T cells was isolated using the Quick-RNA kit (Zymo Research, Irvine, CA, USA). Reverse transcription was performed using a High-Capacity cDNA Reverse Transcription Kit (cat. no. 4368814, Thermo Fisher Scientific, Waltham, MA, USA). Quantitative PCR was performed with the SsoAdvanced Universal SYBR® Green Supermix (cat. no. 1725270, Bio-Rad, Hercules, CA, USA). The Eppendorf Realplex Mastercycler (Eppendorf GmbH) was set to the following amplification parameters: 30 s at 95 °C and 40 cycles of 15 s at 95 °C followed by 30 s at $X$ °C ($X$ = 59 for insulin, 64 for IL-12, 59 or 64 for GAPDH). The relative threshold cycle ($C_t$) was determined and normalized to the endogenous glyceraldehyde 3-phosphate dehydrogenase (GAPDH) transcript. The fold change for each transcript relative to the control was calculated using the comparative $C_t$ method. qPCR primer pairs are listed in Supplementary Table 4.

**Cell imaging**. Cells were fixed using 2% paraformaldehyde in phosphate-buffered saline (PBS) for 20 min. Microscope images were taken using a Nikon Ti-U widefield inverted microscope equipped with a Nikon Intensilight. Green fluorescence was recorded using a 469/35 nm excitation filter, 495 nm long pass filter, and 520/35 nm emission filter. Images were processed using the ImageJ package.

**Data representation and statistics**. Representative data are presented for each figure as individual values (black dots) and mean values (bars). $n = 3$ refers to biological replicates. Statistical significance was calculated with a two-tailed $t$-test, using GraphPad Prism 7.04 software. Normal distributions were assumed. Variance equality was tested using the $f$-test and Welch's correction to the $t$-test was applied when the variances of compared data sets were significantly different ($p < 0.05$). Whenever the result of an assay was below the detection limit (BDL), the value "0" was assigned to that sample to calculate statistical significance. The source data is provided as a Source Data file.

**Reporting summary**. Further information on research design is available in the Nature Research Reporting Summary linked to this article.

## Data availability
The authors declare that all the data supporting the findings of this study are available within the paper and its supplementary information files. Sequence data of original plasmids have been deposited in GenBank under accession codes MN811100 [https://www.ncbi.nlm.nih.gov/nuccore/MN811100]—MN811119 [https://www.ncbi.nlm.nih.gov/nuccore/MN811119] and original plasmids are available upon request. All vector information is provided in Supplementary Table 1. Numerical data and detailed statistical analysis for each figure is provided in Supplementary Table 5. Source data is provided in the Source Data file.

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

## Acknowledgements

We thank Viktor Hällman for sharing figure templates, Gina Melchner von Dydiowa for sharing plasmids, Pratik Saxena for his advice and helpful discussions, Erik Aznauryan for sharing Jurkat cells.

## Author contributions

K.K. and L.S. designed the project. K.K., L.S. and M.F. wrote the manuscript, K.K., L.S. and H.K. designed and performed the experiments. K.K., L.S. and M.F. analyzed the results.

## Competing interests

The authors declare no competing interests.
