## [Peer Review File · Nature Communications]

Reviewers' Comments:

Reviewer #1:

Remarks to the Author:

In their revision of the paper, the authors showed that they understood my concerns and addressed them satisfactorily.

In particular, the authors emphasized pathway-dependence of GEARS in the text and evaluated GEARNFAT effects on other NFAT-dependent gene.

The authors compared their results with versions of MESA receptor (Fig. S5) and increased clarity of the physiological concentration of inducers, as well as performing dose-response experiments for all 4 GEARS (Fig. S11)

Importantly, the authors showed applications of the system in other cell lines: human mesenchymal cells and Jurkat, which definitively increase the impact of the paper.

Overall, I am very satisfied with the revision and think that it increased the clarity and the relevance of the paper, which I now consider ready for publication.

Leonardo Morsut

Reviewer #2:

Remarks to the Author:

In the revised version of the manuscript, Krawczyk et al addressed the points raised and substantially improved it.

In general, I find it suitable for a publication on Nature Communications after commenting on one possible issue on the system's applicability.

One important comment was made by the reviewers is how GEAR affects endogenous signaling pathway in the OFF state. As shown in SF6, NFAT-dependent genes are downregulated in Jurkat stable cell line.

Given previous works on dCas9 transcription systems (i.e. Kiani et al Nat Methods 2015), dCas9 complexed with gRNA can repress promoters by steric hindrance, therefore this result is not surprising. However, this may affect the general applicability of GEARS.

The authors should comment on this aspect, considering both advantages and disadvantages and what could be a real-setting application.

Reviewer #3:

Remarks to the Author:

The revised manuscript added new data to support the GEAR system can rewire inputs to different outputs. The authors also compared the GEAR system with alternative systems (MESA). This system showed unspecific effects on native genes downstream of these transcription factors. There are questions related to the data itself:

1. In Fig 2 for transgenes and Fig 3 for genomic targets, how much target gene activation is caused by the presence of input signals that interacts with the native host transcription factors?

2. In all figures, what are controls (labelled with C)? Are the control 'without the input signal' or 'without the GEAR constructs'? Both controls are probably needed to draw conclusions that their system is inducible upon adding the inputs.

3. The paper should be rewritten to be more concise and to avoid overclaims about potential

applications. It is also strange to talk about Fig 4 ahead of Fig 2e-l.

4. In Fig S2, how do authors explain the spontaneous activation of the control samples over time?

5. The authors didn't describe how they obtained the guide RNAs that were used for endogenous genes. Are these obtained by screening across many guides or reported somewhere else? They should include this information.

Reviewer #1 (Remarks to the Author):

In their revision of the paper, the authors showed that they understood my concerns and addressed them satisfactorily.

In particular, the authors emphasized pathway-dependence of GEARS in the text and evaluated GEARNFAT affects on other NFAT-dependent gene.

The authors compared their results with versions of MESA receptor (Fig. S5) and increased clarity of the physiological concentration of inducers, as well as performing dose-response experiments for all 4 GEARS (Fig. S11)

Importantly, the authors showed applications of the system in other cell lines: human mesenchymal cells and Jurkat, which definitively increase the impact of the paper.

Overall, I am very satisfied with the revision and think that it increased the clarity and the relevance of the paper, which I now consider ready for publication.

Leonardo Morsut

Dear Prof. Morsut, we are glad that we could meet your expectations and appreciate your favourable opinion. We thank you for the constructive suggestions that helped to improve this manuscript.

Reviewer #2 (Remarks to the Author):

In the revised version of the manuscript, Krawczyk et al addressed the points raised and substantially improved it.

In general, I find it suitable for a publication on Nature Communications after commenting on one possible issue on the system' applicability.

We appreciate your positive opinion and acknowledge the useful suggestions regarding control experiments.

One important comment was made by the reviewers is how GEAR affects endogenous signaling pathway in the OFF state. As shown in SF6, NFAT-depend genes are downregulated in Jurkat stable cell line.

Given previous works on dCas9 transcription systems (i.e. Kiani et al Nat Methods 2015), dCas9 complexed with gRNA can repress promoters by steric hindrance, therefore this result is not surprising. However, this may affect the general applicability of GEARS.

The authors should comment on this aspect, considering both advantages and disadvantages and what could be a real-setting application.

We would like to thank the reviewer for pointing that out. It was shown in Kiani et al., 2015, Nat Methods 12, 1051–1054, that the binding of a Cas9-transactivator fusion protein downstream of

the TATA-box of the promoter region can in some cases inhibit transgene expression. The authors tested 8 experimental variants and 1 of them was capable of inhibition. A previous publication by the same first author (Kiani et al. 2014, Nature Methods 11,723–726) showed that such an inhibitory effect is more robust in the absence of a transactivator. However, other work indicates that steric hindrance might also activate gene expression by interfering with natural transcription factor-DNA interactions (Shariati et al. 2019, Molecular Cell 74, 622–633).

In our setting the guide RNA was designed to bind upstream of the transcription site, and we did not observe inhibition due to dCas9 binding, as shown in **Supplementary Fig. 10** (no observed difference between no GEAR/gRNA_{TGFβ} vs no GEAR/gRNA_{INS}). The observed downregulation of endogenous genes shown in **Supplementary Fig. 6** is unspecific and may be caused by a bias of the signalling pathways towards GEARs. This is similar to the squelching effects seen in engineered signalling cascades (Atwood et al., 2011, BMC Genomics 12:14; Tubio et al., 2010, JBC 285:14990) or synthetic transcription factors (Sadowski et al., 1988, Nature 335: 563), which may titrate away critical components of the signalling and transcription machineries, respectively, and so may lead to decreased expression capacity of some target genes. We have confirmed this inherent effect by an additional control experiment showing that increasing the amount of the GEAR_{NFAT}-encoding plasmid in the transfection mixture increased the GEAR-dependent SEAP production, but decreased NanoLuc luciferase production controlled by another NFAT-dependent promoter (**Supplementary Fig. 7**).

We revised the discussion to increase clarity. We also added a comment on unspecific effects of GEARs, as well as discussing advantages and disadvantages, and their impact on real-setting applications.

Reviewer #3 (Remarks to the Author):

The revised manuscript added new data to support the GEAR system can rewire inputs to different outputs. The authors also compared the GEAR system with alternative systems (MESA). This system showed unspecific effects on native genes downstream of these transcription factors. There are questions related to the data itself:

We are happy that we could address your previous concerns, which helped to improve the manuscript. In this round we hope we have resolved all the remaining questions and comments.

1. In Fig 2 for transgenes and Fig 3 for genomic targets, how much target gene activation is caused by the presence of input signals that interacts with the native host transcription factors?

GEARs rely on endogenous signalling pathways, which normally only activate their native targets. We agree that it is reasonable to expect activation of promoters due to binding of host transcription factors. To evaluate that effect, we included control experiments performed with and without the presence of GEARs. **Fig. S9** presents the control experiments for transgenes (shown in **Fig. 2**). Only a minor activation is present without specific GEARs and guide RNA. To

provide a comprehensive response to this question, we performed a new control experiment for genomic targets (experiments shown in **Fig. 3**). We present the results in new **Supplementary Fig. 10**.

We also revised the related portion of the manuscript to improve the clarity.

2. In all figures, what are controls (labelled with C)? Are the control 'without the input signal' or 'without the GEAR constructs'? Both controls are probably needed to draw conclusions that their system is inducible upon adding the inputs.

We have revised the figures in line with this suggestion.

3. The paper should be rewritten to be more concise and to avoid overclaims about potential applications.

We have taken great care to make the manuscript as concise as possible and to more clearly emphasize the experimental foundation of every claim. In the sections where we speculate about potential applications, we have especially made sure that our word choice clearly shows the speculative nature of these statements. Please also refer to our response to the comment from reviewer 2, where we added additional discussion on "advantages and disadvantages and what could be a real-setting application".

It is also strange to talk about Fig 4 ahead of Fig 2e-l.

We appreciate the concern, as it is common practice to number figures according to their first appearance in the text. We followed this practice for all of the figures as a whole, but decided to make an exception for a few single panels, where we felt this would enhance clarity. We think the current setup enables us to provide logically more consistent and informative figures. We would appreciate the editor's opinion on this matter.

4. In Fig S2, how do authors explain the spontaneous activation of the control samples over time?

Basal expression in the non-induced state is a typical phenomenon inherent to all transgene expression systems. Non-limiting examples include:

Liu et al. 2018, Cell 174(2):259-270.e11

Sedlmayer et al. 2018, Nat Commun. 9(1):1822

Scheller et al. 2018, Nat Chem Biol. 14(7):723-729.

Chassin et al. 2017, Nat Commun. 8(1):1101

Xie et al. 2016, Science 354(6317):1296-1301

Schwarz et al. 2017, Nat Chem Biol. 13(2):202-209

The magnitude of leakiness differs between systems and depends on the activity of the signalling pathways and promoter design. We have emphasized this in the revised version of the manuscript.

5. The authors didn't describe how they obtained the guide RNAs that were used for endogenous genes. Are these obtained by screening across many guides or reported somewhere else? They should include this information.

Thank you for pointing this out. We included the requested information in the methods section of the revised version of the manuscript.

Reviewers' Comments:

Reviewer #3:

Remarks to the Author:

The authors have addressed all my comments. The manuscript is ready for publication.

Reviewer #3 (Remarks to the Author):

The authors have addressed all my comments. The manuscript is ready for publication.

We are glad that we managed to address all issues. Thank you for working with us on improving this manuscript.